# Towards a testable notion of generalisation for generative adversarial networks

## Abstract

We consider the question of how to assess generative adversarial networks, in particular with respect to whether or not they generalise beyond memorising the training data. We propose a simple procedure for assessing generative adversarial network performance based on a principled consideration of what the actual goal of generalisation is. Our approach involves using a test set to estimate the Wasserstein distance between the generative distribution produced by our procedure, and the underlying data distribution. We use this procedure to assess the performance of several modern generative adversarial network architectures. We find that this procedure is sensitive to the choice of ground metric on the underlying data space, and suggest a choice of ground metric that substantially improves performance. We finally suggest that attending to the ground metric used in Wasserstein generative adversarial network training may be fruitful, and provide a concrete formulation for doing so.

## 1 Introduction

Generative adversarial networks (GANs) (Goodfellow et al., 2014) have attracted significant interest as a means for generative modelling. However, recently concerns have been raised about their ability to generalise from training data and their capacity to overfit (Arora & Zhang, 2017; Arora et al., 2017). Moreover, techniques for evaluating the quality of GAN output are either ad hoc, lack theoretical rigor, or are not suitably objective – often times "visual inspection" of samples is the main tool of choice for the practitioner. More fundamentally, it is sometimes unclear exactly what we want a GAN to do: what is the learning task that we are trying to achieve?

In this paper, we provide a simple formulation of the GAN training framework, which consists of using a finite dataset to estimate an underlying data distribution. The quality of GAN output is measured precisely in terms of a statistical distance $D$ between the estimated and true distribution. Within this context, we propose an intuitive notion of what it means for a GAN to generalise.

We also show how our notion of performance can be measured empirically for any GAN architecture when $D$ is chosen to be a Wasserstein distance, which – unlike other methods such as the inception score (Salimans et al., 2016) – requires no density assumptions about the data-generating distribution. We investigate this choice of $D$ empirically, finding that its performance is heavily dependent on the choice of ground metric underlying the Wasserstein distance. We suggest a novel choice of ground metric that we show performs well, and also discuss how we might otherwise use this observation to improve the design of Wasserstein GANs (WGANs) (Arjovsky et al., 2017).

## 2 The objective of generative modelling

GANs promise a means for learning complex probability distributions in an unsupervised fashion. In order to assess their effectiveness, we must first define precisely what we mean by this. We seek to do so in this section, presenting a formulation of the broader goal of generative modelling that we believe is widely compatible with much present work in this area. We also provide a natural notion of generalisation that arises in our framework.

Our setup consists of the following components. We assume some distribution $\pi$ on a set $\mathcal{X}$. For instance, $\mathcal{X}$ may be the set of 32x32 colour images, and $\pi$ the distribution from which the CIFAR-10

dataset was sampled. We assume that $\pi$ is completely intractable: we do not know its density (or even if it has one), and we do not have a procedure to draw novel samples from it. However, we do suppose that we have a fixed dataset $X$ consisting of samples $x_1, \cdots, x_{|X|} \overset{\text{iid}}{\sim} \pi$. Equivalently, we have the empirical distribution

$$\hat{X} := \frac{1}{|X|} \sum_{n=1}^{|X|} \delta_{x_n},$$

where $\delta$ denotes the Dirac mass.

Let $\mathcal{P}(\mathcal{X})$ denote the set of probability distributions on $\mathcal{X}$. Our aim is to use $X$ to produce a distribution in $\mathcal{P}(\mathcal{X})$ that is as "close" as possible to $\pi$. We choose to measure closeness in terms of a function $D : \mathcal{P}(\mathcal{X}) \times \mathcal{P}(\mathcal{X}) \to \mathbb{R}$. Usually $D$ will be chosen to be a *statistical divergence*, which means that $D(P, Q) \geq 0$ for all $P$ and $Q$, with equality iff $P = Q$. The task of a learning algorithm $\alpha$ in this context is then as follows:

$$\boxed{\text{Select } \alpha(X) \in \mathcal{P}(\mathcal{X}) \text{ such that } D(\alpha(X), \pi) \text{ is as small as possible.}} \tag{1}$$

We believe (1) constitutes an intuitive and useful formulation of the problem of generative modelling that is largely in keeping with present research efforts.

Now, we can immediately see that one possibility is simply to choose $\alpha(X) := \hat{X}$. Moreover, in the case that $D$ is a metric for the weak topology on $\mathcal{P}(\mathcal{X})$ such as a Wasserstein distance, we have that $D(\hat{X}, \pi) \to 0$ almost surely as $|X| \to \infty$, so that, assuming $|X|$ is large enough, we can already expect $D(\hat{X}, \pi)$ to be small. This then suggests the following natural notion of generalisation:

$$\boxed{\text{A choice of } \alpha \text{ generalises for a given } X \text{ if } D(\alpha(X), \pi) < D(\hat{X}, \pi).} \tag{2}$$

In other words, using $\alpha$ here has actually achieved something: perhaps through some process of smoothing or interpolation, it has injected additional information into $\hat{X}$ that has moved us closer to $\pi$ than we were *a priori*.

## 3 GENERALISATION IN GANS

The previous section presented (1) as a general goal of generative modelling. In this section, we turn specifically to GANs. We begin by providing a general model for how many of the existing varieties of GAN operate, at least ideally. We then show how this model fits into our framework above, before considering the issue of generalisation in this context.

Most GAN algorithms in widespread use adhere to the following template: they take as input a distribution $P$, from which we assume we can sample, and compute (or approximate)

$$\Gamma(P) := \underset{Q \in \mathcal{Q}}{\arg\min}\, D_\Gamma(P, Q)$$

for some choices of $\mathcal{Q} \subseteq \mathcal{P}(\mathcal{X})$ and $D_\Gamma : \mathcal{P}(\mathcal{X}) \times \mathcal{P}(\mathcal{X}) \to \mathbb{R}$. In other words, in the ideal case, a GAN maps $P$ to its $D_\Gamma$-projection onto $\mathcal{Q}$. Note that we will not necessarily have that $D_\Gamma = D$: $D_\Gamma$ is fixed given a particular GAN architecture, whereas the choice of $D$ is simply a feature of our problem definition (1) and is essentially ours to make.

In practice, $\mathcal{Q}$ is the set of pushforward measures $\nu \circ G^{-1}$ obtained from a fixed noise distribution $\nu$ on a noise space $\mathcal{Z}$ and some set $\mathcal{G}$ of functions $G : \mathcal{Z} \to \mathcal{X}$. Precisely, $\mathcal{Q} = \left\{ \nu \circ G^{-1} : G \in \mathcal{G} \right\}$. $\mathcal{G}$ itself usually corresponds to the set of functions realisable by some neural network architecture, and $\nu$ is some multivariate uniform or Gaussian distribution. However, numerous choices of $D_\Gamma$ have been proposed: the original GAN formulation (Goodfellow et al., 2014) took $D_\Gamma$ to be the Jenson-Shannon divergence, whereas the $f$-GAN (Nowozin et al., 2016) generalised this to arbitrary $f$-divergences, and the Wasserstein GAN (Arjovsky et al., 2017) advocated the Wasserstein distance. Many of the results proved in these papers involve showing that (usually under some assumptions, such as sufficient discriminator capacity) a proposed objective for $G$ is in fact equivalent to $D_\Gamma(P, \nu \circ G^{-1})$.

In terms of our framework in the previous section, using a GAN $\Gamma$ amounts to choosing

$$\alpha(X) := \Gamma(\hat{X}) = \arg\min_{Q \in \mathcal{Q}} D_{\Gamma}(\hat{X}, Q).$$

We emphasise again the important distinction between $D$ and $D_{\Gamma}$. In our setup, minimising $D$ defines our ultimate goal, whereas minimising $D_{\Gamma}$ (over $\mathcal{Q}$) defines how we will attempt to achieve that goal. Even if $D \neq D_{\Gamma}$, it is still at least conceivable that $D(\Gamma(\hat{X}), \pi)$ might be small, and therefore this choice of $\alpha$ might be sensible. Also note that, crucially, $\Gamma$ receives $\hat{X}$ as input rather than $\pi$ itself. We only have access to a fixed number of CIFAR-10 samples, for example, not an infinite stream. Moreover, training GANs usually involves making many passes over the same dataset, so that, in effect, sampling from $P$ will repeatedly yield the same data points. We would not expect this to occur with nonzero probability if $P = \pi$ for most $\pi$ of interest.

The observation that $P$ is $\hat{X}$ rather than $\pi$ was also recently made by Arora et al. (2017). The authors argue that this introduces a problem for the ability of GANs to generalise, since, if $D_{\Gamma}$ is a divergence (which is almost always the case), and if $\mathcal{Q}$ is too big (in particular, if it is big enough that $\hat{X} \in \mathcal{Q}$), then we trivially have that $\Gamma(\hat{X}) = \hat{X}$. In other words, the GAN objective appears actively to encourage $\Gamma(\hat{X})$ to memorise the dataset $X$, and never to produce novel samples from outside of it. The authors' proposed remedy involves trying to find a better choice of $D_{\Gamma}$. The problem, they argue, is that popular choices of $D_{\Gamma}$ do not satisfy the condition

$$D_{\Gamma}(\pi, Q) \approx D_{\Gamma}(\hat{X}, Q) \text{ with high probability given a modest number of samples in } X. \quad (3)$$

They point out that this is certainly violated when $D_{\Gamma}$ is the Jensen-Shannon divergence JS, since

$$\mathrm{JS}(P \parallel Q) = \log 2$$

when one of $P$ and $Q$ is discrete and the other continuous, and give a similar result for $D_{\Gamma}$ a Wasserstein distance in the case that $\pi$ is Gaussian. As a solution, they introduce the *neural network distance* $D_{\mathrm{NN}}$ defined by

$$D_{\mathrm{NN}}(P, Q) := \max_{f \in \mathcal{F}} \mathbb{E}_{x \sim P}\left[f(x)\right] - \mathbb{E}_{x \sim Q}\left[f(x)\right]$$

for some choice of a class of functions $\mathcal{F}$. They show that, assuming some smoothness conditions on the members of $\mathcal{F}$, the choice $D_{\Gamma} = D_{\mathrm{NN}}$ satisfies (3), which means that if we minimise $D_{\mathrm{NN}}(\hat{X}, Q)$ in $Q$ then we can be confident that the value of $D_{\mathrm{NN}}(\pi, Q)$ is small also.

However, we do not believe that (3) is sufficient to ensure good generalisation behaviour for GANs. What we care about ultimately is not the value of $D_{\Gamma}$, but rather of $D$, and (3) invites choosing $D_{\Gamma}$ in such a way that gives no guarantees about $D$ at all. We see, for instance, that the degenerate choice $D_0(P, Q) := 0$ trivially satisfies (3), and indeed is also a pseudometric, just like $D_{\mathrm{NN}}$. It is therefore unclear what mathematical properties of $D_{\mathrm{NN}}$ render it more suitable for estimating $\pi$ than the obviously undesirable $D_0$. The authors do acknowledge this shortcoming of $D_{\mathrm{NN}}$, pointing out that $D_{\mathrm{NN}}(P, Q)$ can be small even if $P$ and $Q$ are "quite different" in some sense.

The problematic consequences of the fact that $P = \hat{X}$ apply only in the case that $\mathcal{Q}$ is too large. In practice, however, $\mathcal{Q}$ is heavily restricted, since $\mathcal{G}$ is restricted via a choice of neural network architecture; hence we do not know *a priori* whether $\Gamma(\hat{X}) = \hat{X}$ is even possible. As such, we do not see the choice of $\alpha(X) = \Gamma(\hat{X})$ as necessarily a bad idea, and instead believe that it is an open empirical question as to how well GANs perform the task (1). In fact, this $\alpha$ falls perfectly within the framework of minimum distance estimation (Wolfowitz, 1957; Basu et al., 2011), which involves estimating an underlying distribution by minimising a distance measure to a given empirical distribution.

## 4 Testing GANs

Our goal in this section is to assess how well GANs achieve (1) by estimating $D(\Gamma(X), \pi)$ for various $\Gamma$ and $\pi$. This raises some difficulties, given that $\pi$ is intractable. Our approach is to take $D$ to be the first Wasserstein distance $W_{d_{\mathcal{X}}}$ defined by

$$W_{d_{\mathcal{X}}}(P, Q) = \inf_{\gamma \in \Pi(P, Q)} \int_{\mathcal{X} \times \mathcal{X}} d_{\mathcal{X}}(x, y) \, \mathrm{d}\gamma(x, y),$$

where $d_{\mathcal{X}}$ is a metric on $\mathcal{X}$ referred to as the *ground metric*, and $\Pi(P, Q)$ denotes the set of joint distributions on the product space $\mathcal{X} \times \mathcal{X}$ with marginals $P$ and $Q$. The Wasserstein distance is appealing since it is sensitive to the topology of the underlying set $\mathcal{X}$, which we control by our choice of $d_{\mathcal{X}}$. Moreover, $W_{d_{\mathcal{X}}}$ metricises weak convergence for the Wasserstein space $\mathcal{P}_{d_{\mathcal{X}}}(\mathcal{X})$ defined by

$$\mathcal{P}_{d_{\mathcal{X}}}(\mathcal{X}) := \left\{ P \in \mathcal{P}(\mathcal{X}) : \int_{\mathcal{X}} d_{\mathcal{X}}(x_0, y) \, \mathrm{d}P(y) < \infty \text{ for some } x_0 \in \mathcal{X} \right\}$$

(see (Villani, 2008)). Consequently, if we denote by $A$ a set of samples $a_1, \cdots, a_{|A|} \overset{\mathrm{iid}}{\sim} \alpha(X)$, and by $Y$ a set of samples (separate from $X$) $y_1, \cdots, y_{|Y|} \overset{\mathrm{iid}}{\sim} \pi$, with $\hat{A}$ and $\hat{Y}$ the corresponding empirical distributions, then, provided

$$\alpha(X), \pi \in \mathcal{P}_{d_{\mathcal{X}}}(\mathcal{X}), \tag{4}$$

we have that $D(\hat{A}, \hat{Y}) \to D(\alpha(X), \pi)$ almost surely as $\min\{|A|, |Y|\} \to \infty$. Note that condition (4) holds automatically in the case that $(\mathcal{X}, d_{\mathcal{X}})$ is compact, since then $\mathcal{P}_{d_{\mathcal{X}}}(\mathcal{X}) = \mathcal{P}(\mathcal{X})$.

As such, to estimate $D(\alpha(X), \pi)$, for $D = W_{d_{\mathcal{X}}}$, we propose the following. Before training, we move some of our samples from $X$ into a testing set $Y$. We next train our GAN on $X$, obtaining $\alpha(X)$. We then take samples $A$ from $\alpha(X)$, and obtain the estimate

$$W_{d_{\mathcal{X}}}(\hat{A}, \hat{Y}) \approx W_{d_{\mathcal{X}}}(\alpha(X), \pi),$$

where the left-hand side can be computed exactly by solving a linear program since both $\hat{A}$ and $\hat{Y}$ are discrete (Villani, 2003). We can also use the same methodology to estimate $W_{d_{\mathcal{X}}}(\hat{X}, \pi)$ by $W_{d_{\mathcal{X}}}(\hat{X}, \hat{Y})$, which suggests testing if

$$W_{d_{\mathcal{X}}}(\hat{A}, \hat{Y}) < W_{d_{\mathcal{X}}}(\hat{X}, \hat{Y}). \tag{5}$$

as a proxy for determining whether (2) holds. A summary of our procedure is given in Algorithm 1.

---

**Algorithm 1** Procedure for testing GANs

---

1: Split samples from $\pi$ into a training set $X$ and a testing set $Y$
2: Compute $\alpha(X)$ by training a GAN on $X$
3: Obtain a sample $A$ from $\alpha(X)$
4: Approximate
$$W(\alpha(X), \pi) \approx W(\hat{A}, \hat{Y}),$$
   where the right-hand side can be computed by solving a linear program
5: Similarly, test whether
$$W(\hat{A}, \hat{Y}) < W(\hat{X}, \hat{Y})$$
   as a proxy for $W(\alpha(X), \pi) < W(A, \pi)$

---

### 4.1 RESULTS

We applied our methodology to test two popular GAN varieties – the DCGAN (Radford et al., 2015) and the Improved Wasserstein GAN (I-WGAN) (Gulrajani et al., 2017) – on the MNIST and CIFAR-10 datasets. In all cases when computing the relevant Wasserstein distances, our empirical distributions consisted of 10000 samples.

#### 4.1.1 $L^2$ AS GROUND METRIC

We initially took our ground metric $d_{\mathcal{X}}$ to be the $L^2$ distance. This has the appealing property of making $(\mathcal{X}, d_{\mathcal{X}})$ compact when $\mathcal{X}$ is a space of RGB or greyscale images, therefore ensuring that (4) holds.

Figure 1 shows $W_{L^2}(\hat{A}, \hat{Y})$ plotted over multiple training runs for MNIST and CIFAR-10 when $A$ is obtained from an I-WGAN, with typical samples shown in Figures 6 and 7 in the appendix. For

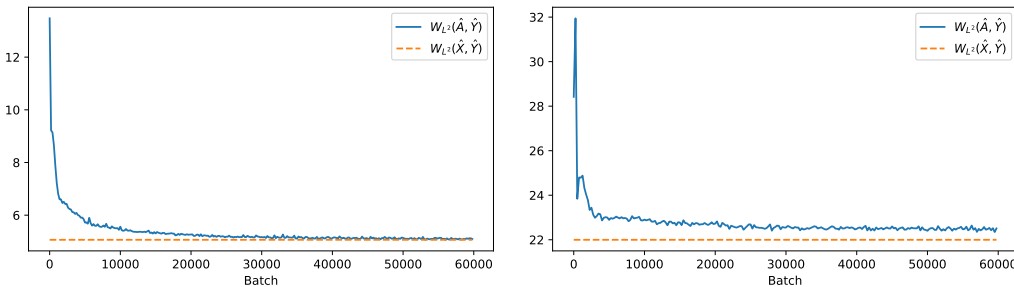

Figure 1: Output of Algorithm 1 with $d_{\mathcal{X}} = L^2$ for I-WGAN trained on MNIST (left) and CIFAR-10 (right)

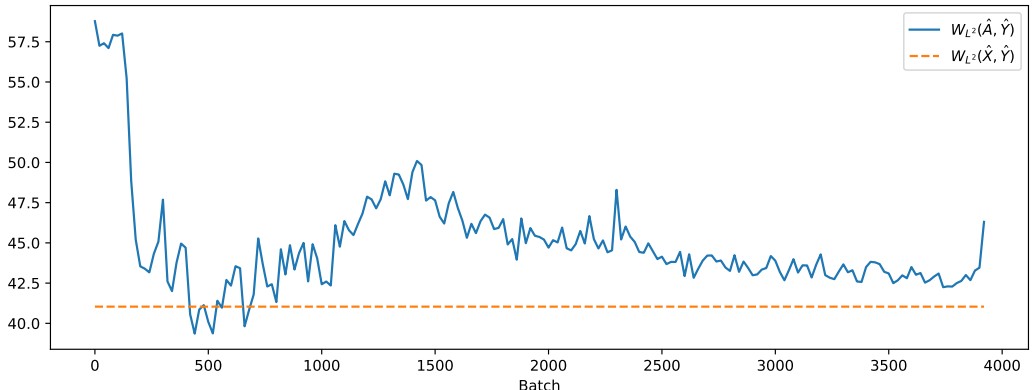

Figure 2: Output of Algorithm 1 with $d_{\mathcal{X}} = L^2$ for DCGAN trained on CIFAR-10

both datasets, $W_{L^2}(\hat{A}, \hat{Y})$ decays towards an asymptote in a way that nicely corresponds to the visual quality of the samples produced. Moreover, $W_{L^2}(\hat{A}, \hat{Y})$ is much closer to $W_{L^2}(\hat{X}, \hat{Y})$ towards the tail end of training for MNIST than for CIFAR-10. This seems to reflect the fact that, visually, the eventual I-WGAN MNIST samples do seem quite close to true MNIST samples, whereas the eventual I-WGAN CIFAR-10 samples are easily identified as fake.

However, when we re-ran the same experiment using a DCGAN on CIFAR-10, we obtained the $W_{L^2}(\hat{A}, \hat{Y})$ trajectories shown in Figure 2. Typical examples are shown in Figure 8. Strangely, we observe that $W_{L^2}(\hat{A}, \hat{Y}) < W_{L^2}(\hat{X}, \hat{Y})$ very early on in training – at around batch 500 – when the samples resemble the heavily blurry Figure 8b. This raises some obvious concerns about the appropriateness of $W_{L^2}$ as a metric for GAN quality.

We therefore sought to understand this strange behaviour. Motivated by the visual blurriness of the samples in Figure 8b, we explored the effect on $W_{L^p}(\hat{X}, \hat{Y})$ of blurring the CIFAR-10 training set $X$. In particular, we let $X$ and $Y$ each consist of 10000 distinct CIFAR-10 samples in $X$. We then independently convolved each channel of each image with a Gaussian kernel having standard deviation $\sigma$, obtaining a blurred dataset $\beta_\sigma(X)$ and corresponding empirical distribution $\hat{\beta}_\sigma(X)$. The visual effect of this procedure is shown in Figure 9 in the appendix. We then computed $W_{L^p}(\hat{\beta}_\sigma(X), \hat{Y})$ with $\sigma$ ranging between 0 and 10 for a variety of values of $p$. The results of this experiment in the case $p = 2$ are shown in Figure 3, and similar results were observed for other values of $p$: in all cases, we found that

$$W_{L^p}(\hat{X}, \hat{Y}) > W_{L^p}(\hat{\beta}_\sigma(X), \hat{Y})$$

whenever $\sigma > 0$. That is, blurring $X$ by any amount brings $\hat{X}$ closer to $\hat{Y}$ in $W_{L^p}$ than not. This occurs even though $X$ is distributed identically to $Y$ (both being drawn from $\pi$), while $\beta_\sigma(X)$ (presumably) is not when $\sigma > 0$.

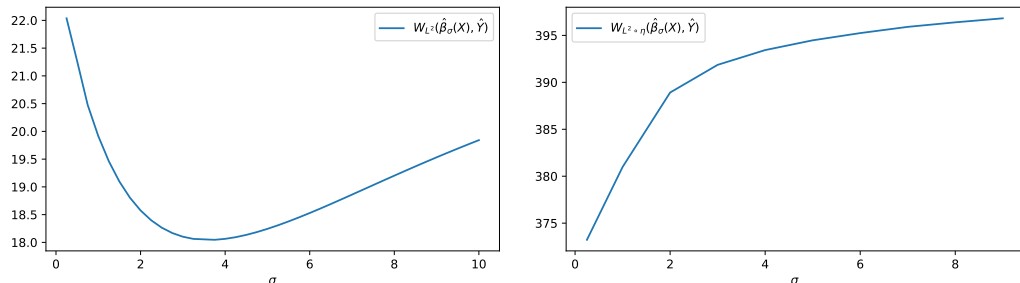

Figure 3: Effect of blurring CIFAR-10 on $W_{L^2}$ (left) and $W_{L^2 \circ \eta}$ (right)

### 4.1.2 EMBEDDED $L^2$ AS GROUND METRIC

To remedy these issues, we sought to replace $L^2$ with a choice of $d_{\mathcal{X}}$ that is more naturally suited to the space of images in question. To this end we tried mapping $\mathcal{X}$ through a fixed pre-trained neural network $\eta$ into a feature space $\mathcal{Y}$, and then computing distances using some metric $d_{\mathcal{Y}}$ on $\mathcal{Y}$, rather than in $\mathcal{X}$ directly. It is easily seen that, provided $\eta$ is injective, $d_{\mathcal{Y}} \circ \eta : \mathcal{X} \times \mathcal{X} \to \mathbb{R}$ defined by

$$(d_{\mathcal{Y}} \circ \eta)(x, x') := d_{\mathcal{Y}}(\eta(x), \eta(x'))$$

is a metric. It also holds that, when $\eta$ is $(d_{\mathcal{X}}, d_{\mathcal{Y}})$-continuous, $(\mathcal{X}, d_{\mathcal{Y}} \circ \eta)$ is compact when $(\mathcal{X}, d_{\mathcal{X}})$ is: given a sequence $x_i \subseteq \mathcal{X}$, there exists a subsequence $x_{i'}$ that converges in $d_{\mathcal{X}}$ to some $x$ (by compactness), so that $(d_{\mathcal{Y}} \circ \eta)(x_{i'}, x) = d_{\mathcal{Y}}(\eta(x_{i'}), \eta(x)) \to 0$ by continuity of $\eta$. Neither of these conditions on $\eta$ – injectivity and $(d_{\mathcal{X}}, d_{\mathcal{Y}})$-continuity – are unreasonable to expect from a typical neural network trained using stochastic gradient descent (at least, when $d_{\mathcal{X}}$ and $d_{\mathcal{Y}}$ are typical metrics such as $L^p$ distances). Consequently, $W_{d_{\mathcal{Y}} \circ \eta}$ constitutes a valid metric on $\mathcal{P}_{d_{\mathcal{Y}} \circ \eta}(\mathcal{X})$, and (4) is automatically satisfied.

To test the performance of $W_{d_{\mathcal{Y}} \circ \eta}$, we repeated the blurring experiment described above. We took $\eta(x)$ to be the result of scaling $x \in \mathcal{X}$ to size 224x224, mapping the result through a DenseNet-121 (Huang et al., 2016) pre-trained on ImageNet (Deng et al., 2009), and extracting features immediately before the linear output layer. Under the same experimental setup as above otherwise, we obtained the plot of $W_{L^2 \circ \eta}(\beta_\sigma(\hat{X}), \hat{Y})$ shown in Figure 3. Happily, we now see that this curve increases monotonically as $\sigma$ grows in accordance with the declining visual quality of $\beta_\sigma(X)$ shown in Figure 9.

Next, we computed $W_{L^2 \circ \eta}(\hat{A}, \hat{Y})$ over the course of GAN training. For the I-WGAN we obtained the results on MNIST and CIFAR-10 shown in Figure 4;[1] for the DCGAN on CIFAR-10, we obtained the curve shown in Figure 5. In all cases we see that $W_{L^2 \circ \eta}(\hat{A}, \hat{Y})$ decreases monotonically towards an asymptote in a way that accurately summarises the visual quality of the samples throughout the training run. Moreover, there is always a large gap between the eventual value of $W_{L^2 \circ \eta}(\hat{A}, \hat{Y})$ and $W_{L^2 \circ \eta}(\hat{X}, \hat{Y})$, which reflects the fact that the GAN samples are still visually distinguishable from real $\pi$ samples. In particular, we see an improvement in this respect for the I-WGAN on MNIST: in Figure 1 the asymptotic value of $W_{L^2}(\hat{A}, \hat{Y})$ for MNIST was barely discernible from $W_{L^2}(\hat{X}, \hat{Y})$, despite the fact that it is still quite easy to tell real samples from generated ones (see e.g. the various mistakes present in Figure 6).

## 5 DISCUSSION AND FUTURE WORK

We believe our work reveals two promising avenues of future inquiry. First, we suggest that $W_{L^p \circ \eta}$ is an appealing choice of $D$, both due to its nice theoretical properties – it metricises weak convergence, and does not require us to make any density assumptions about $\pi$ – and due to its sound empirical performance demonstrated above. It would be very interesting to use this $D$ to produce a systematic and objective comparison of the performance of all current major GAN implementations, and indeed

---

[1]Note that on MNIST in this case we first duplicated each input across three channels for shape compatibility with the DenseNet.

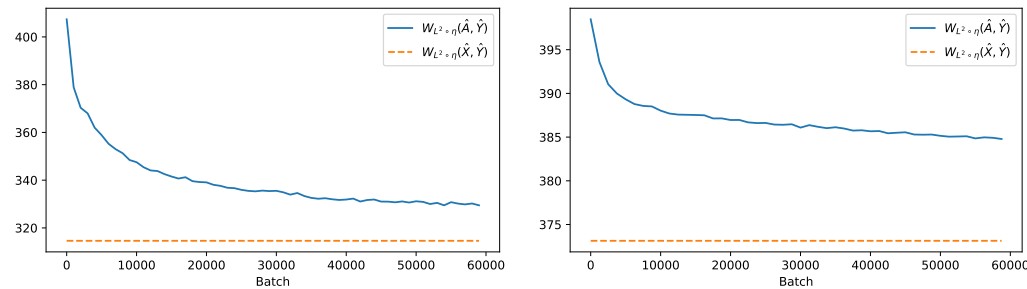

Figure 4: Output of Algorithm 1 with $d_{\mathcal{X}} = L^2 \circ \eta$ for I-WGAN trained on MNIST (left) and CIFAR-10 (right)

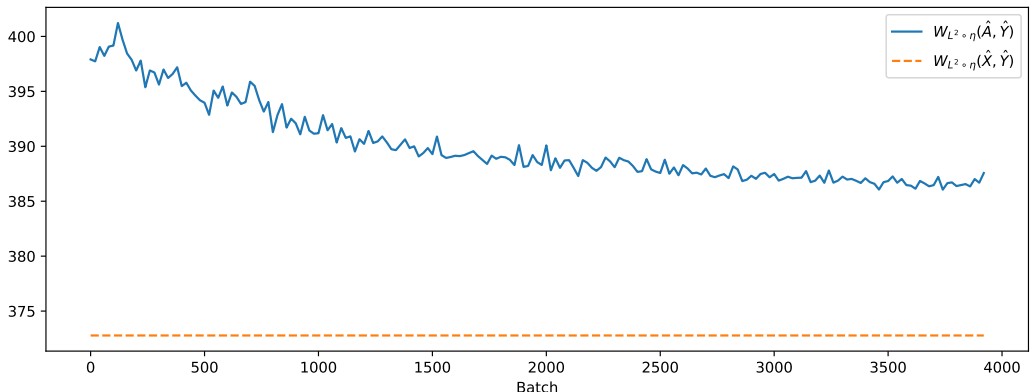

Figure 5: Output of Algorithm 1 with $d_{\mathcal{X}} = L^2 \circ \eta$ for DCGAN trained on CIFAR-10

to use this as a metric for guiding future GAN design. We also view the test (5) as potentially useful for determining whether our algorithms are overfitting. This would be particularly so if applied via a cross-validation procedure: if we consistently observe that (5) holds when training a GAN according to many different $X$ and $Y$ partitions of our total $\pi$ samples, then it seems reasonable to infer that $\alpha(X)$ has indeed learnt something useful about $\pi$.

We also believe that the empirical inadequacy of $W_{L^2}$ that we observed suggests a path towards a better WGAN architecture. At present, WGAN implementations implicitly use $W_{L^2}$ for their choice of $D_\Gamma$. We suspect that altering this to our suggested $W_{L^2 \circ \eta}$ may yield better quality samples. We briefly give here one possible way to do so that is largely compatible with existing WGAN setups. In particular, following Arjovsky et al. (2017), we take

$$D_\Gamma(P, Q) = \max_{f \in \mathcal{F}} \mathbb{E}_{x \sim P} \left[ f(x) \right] - \mathbb{E}_{x \sim Q} \left[ f(x) \right]$$

for a class $\mathcal{F}$ of functions $f : \mathcal{X} \to \mathbb{R}$ that are all $(L^2 \circ \eta, d_{\mathbb{R}})$-Lipschitz for some fixed Lipschitz constant $K$. Here $d_{\mathbb{R}}$ denotes the usual distance on $\mathbb{R}$. To optimise over such an $\mathcal{F}$ in practice, we can require our discriminator $f : \mathcal{X} \to \mathbb{R}$ to have the form $f(x) := h(\eta(x))$, where $h : \mathcal{Y} \to \mathbb{R}$ is $(d_{\mathcal{Y}}, d_{\mathbb{R}})$-Lipschitz, which entails that $f$ is Lipschitz provided $\eta$ is (which is almost always the case in practice). In other words, we compute

$$D_\Gamma(P, Q) = \max_{h \in \mathcal{F}'} \mathbb{E}_{x \sim P} \left[ h(\eta(x)) \right] - \mathbb{E}_{x \sim Q} \left[ h(\eta(x)) \right],$$

where $\mathcal{F}'$ is a class of $(d_{\mathcal{Y}}, d_{\mathbb{R}})$-Lipschitz functions. Optimising over this objective may now proceed as usual via weight-clipping like (Arjovsky et al., 2017), or via a gradient penalty like (Gulrajani et al., 2017). Note that this suggestion may be understood as training a standard WGAN with the initial layers of the discriminator fixed to the embedding $\eta$; our analysis here shows that this is equivalent to optimising with respect to $W_{L^2 \circ \eta}$ instead of $W_{L^2}$. We have begun some experimentation in this area but leave a more detailed empirical inquiry to future work.

It is also clearly important to establish better theoretical guarantees for our method. At present, we have no guarantee that the number of samples in $A$ and $Y$ are enough to ensure that

$$D(\hat{A}, \hat{Y}) \approx D(\alpha(X), \hat{Y})$$

(perhaps with some fixed bias that is fairly independent of $\alpha$, so that it is valid to use the value of $D(\hat{A}, \hat{Y})$ to compare different choices of $\alpha$), or that (5) entails (2) with high probability. We do however note that some recent theoretical work on the convergence rate of empirical Wasserstein estimations (Weed & Bach, 2017) does suggest that it is plausible to hope for fast convergence of $D(\hat{A}, \hat{Y})$ to $D(\alpha(X), \hat{Y})$. We also believe that the convincing empirical behaviour of $W_{L^2 \circ \eta}$ does suggest that it is possible to say something more substantial about our approach, which we leave to future work.

## 6 RELATED WORK

The maximum mean discrepancy (MMD) is another well-known notion of distance on probability distributions, which has been used for testing whether two distributions are the same or not (Gretton et al., 2012) and also for learning generative models in the style of GAN (Li et al., 2015; Dziugaite et al., 2015; Sutherland et al., 2016; Li et al., 2017). It is parameterised by a characteristic kernel $k$, and defines the distance between probability distributions by means of the distance of $k$'s reproducing kernel Hilbert space (RKHS). The MMD induces the same weak topology on distributions as the one of the Wasserstein distance. Under a mild condition, the MMD between distributions $P$ and $Q$ under a kernel $k$ can be understood as the outcome of the following two-step calculation. First, we pushforward $P$ and $Q$ from their original space $\mathcal{X}$ to a Hilbert space $\mathcal{H}$ (isomorphic to $k$'s RKHS) using a feature function $\phi : \mathcal{X} \to \mathcal{H}$ induced by the kernel $k$. Typically, $\mathcal{H}$ is an *infinite*-dimensional space, such as the set $\ell_2$ of square-summable sequences in $\mathcal{R}^\infty$ as in Mercer's theorem. Let $P'$ and $Q'$ be the resulting distributions on the feature space. Second, we compute the supremum of $\mathbb{E}_{P'(X)}[f(X)] - \mathbb{E}_{Q'(X)}[f(X)]$ over all *linear* 1-Lipschitz functions $f : \mathcal{H} \to \mathbb{R}$. The result of this calculation is the MMD between $P$ and $Q$.

This two-step calculation shows the key difference between MMD and our use of Wasserstein distance and neural embedding $\eta$. While the co-domain of the feature function $\phi$ is an *infinite*-dimensional space (e.g. $\ell_2$) in most cases, its counterpart $\eta$ in our setting uses a *finite*-dimensional space as co-domain. This means that the MMD possibly uses a richer feature space than our approach. On the other hand, the MMD takes the supremum over only *linear* functions $f$ among all 1-Lipschitz functions, whereas the Wasserstein distance considers all these 1-Lipschitz functions. These different balancing acts between the expressiveness of features and that of functions taken in the supremum affect the learning and testing of various GAN approaches as observed experimentally in the literature. One interesting future direction is to carry out systematic study on the theoretical and practical implications of these differences.

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

## A  EXAMPLE I-WGAN SAMPLES

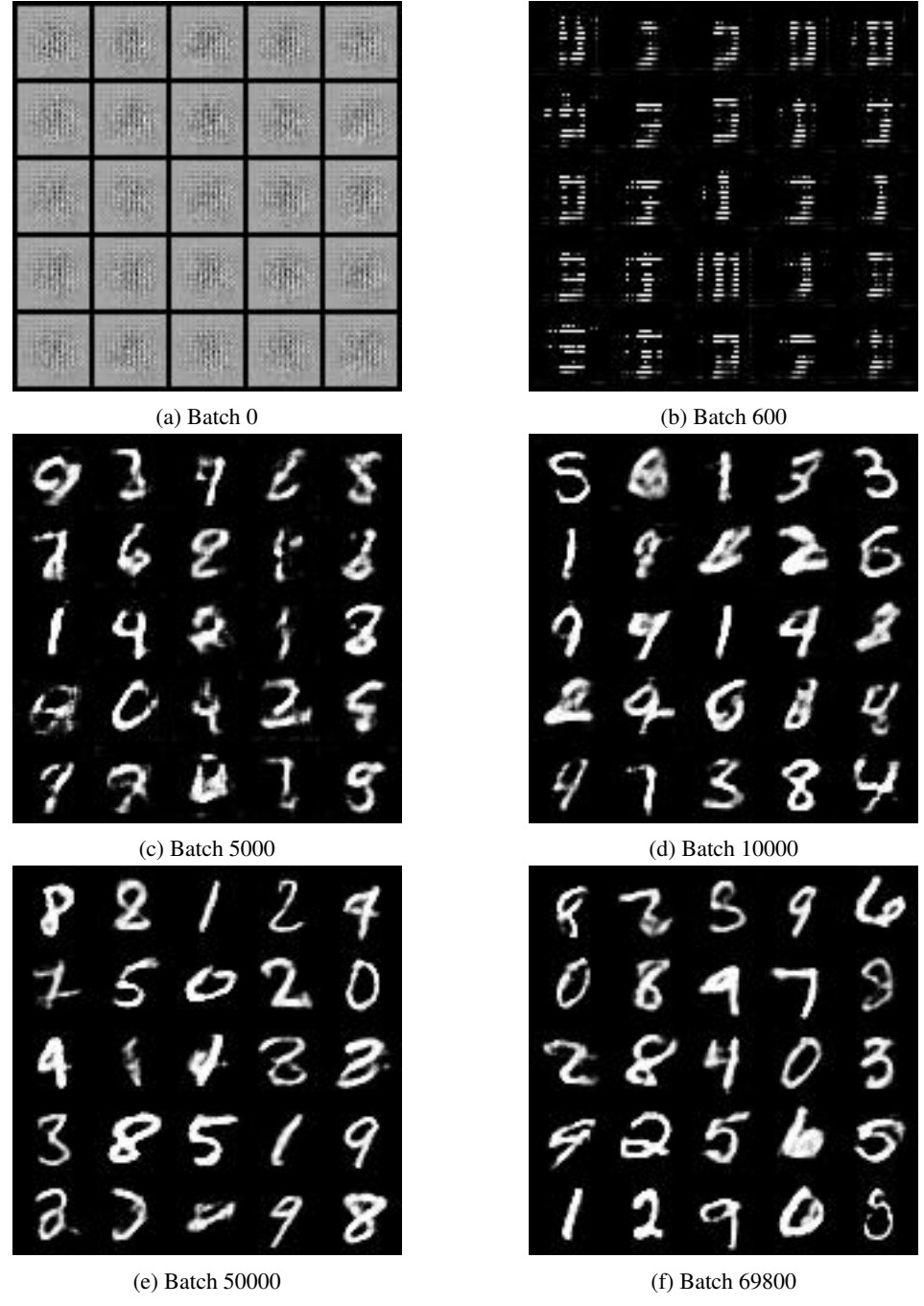

(a) Batch 0

(b) Batch 600

(c) Batch 5000

(d) Batch 10000

(e) Batch 50000

(f) Batch 69800

Figure 6: Samples from I-WGAN trained on MNIST

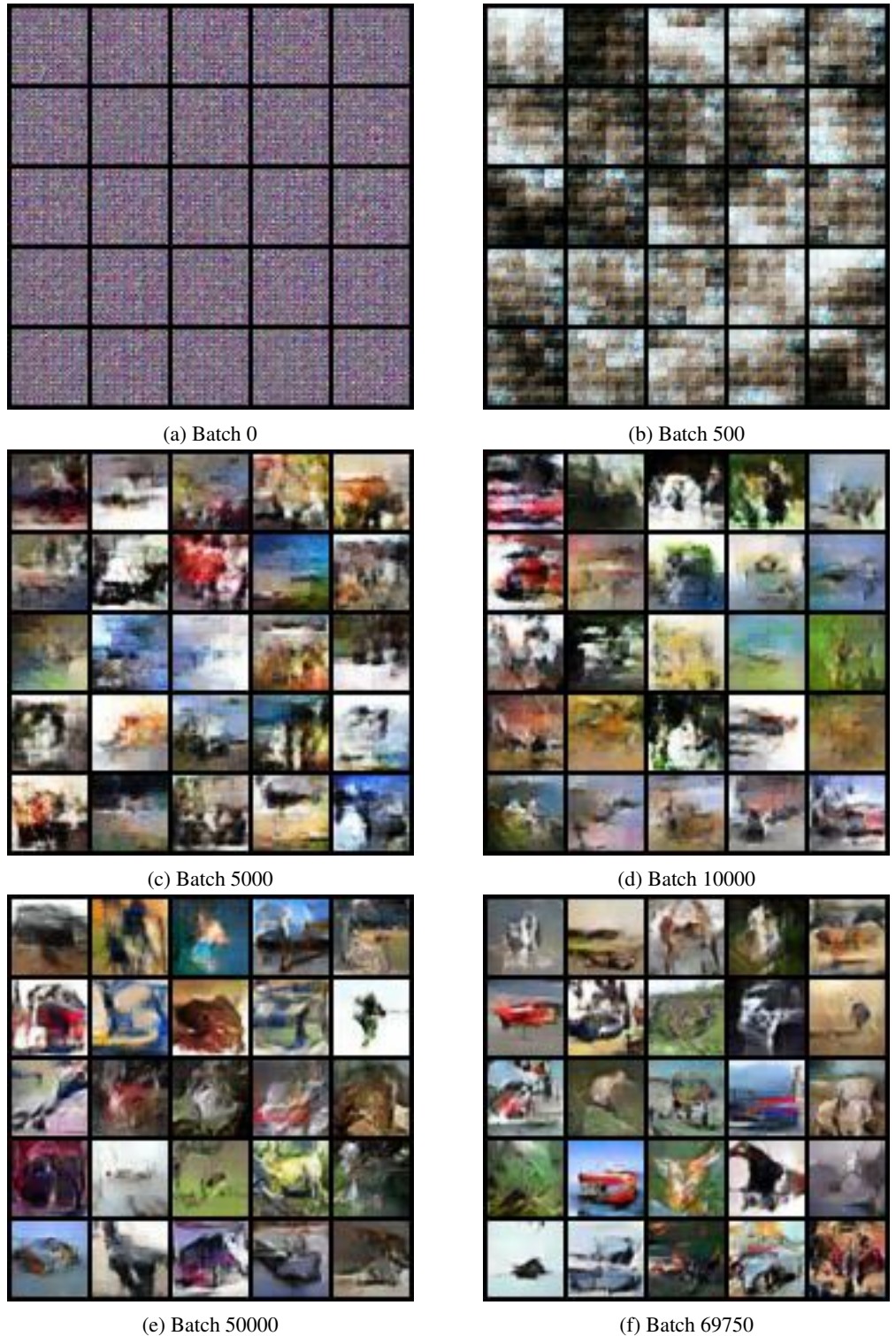

(a) Batch 0

(b) Batch 500

(c) Batch 5000

(d) Batch 10000

(e) Batch 50000

(f) Batch 69750

Figure 7: Samples from I-WGAN trained on CIFAR-10

## B  EXAMPLE DCGAN SAMPLES

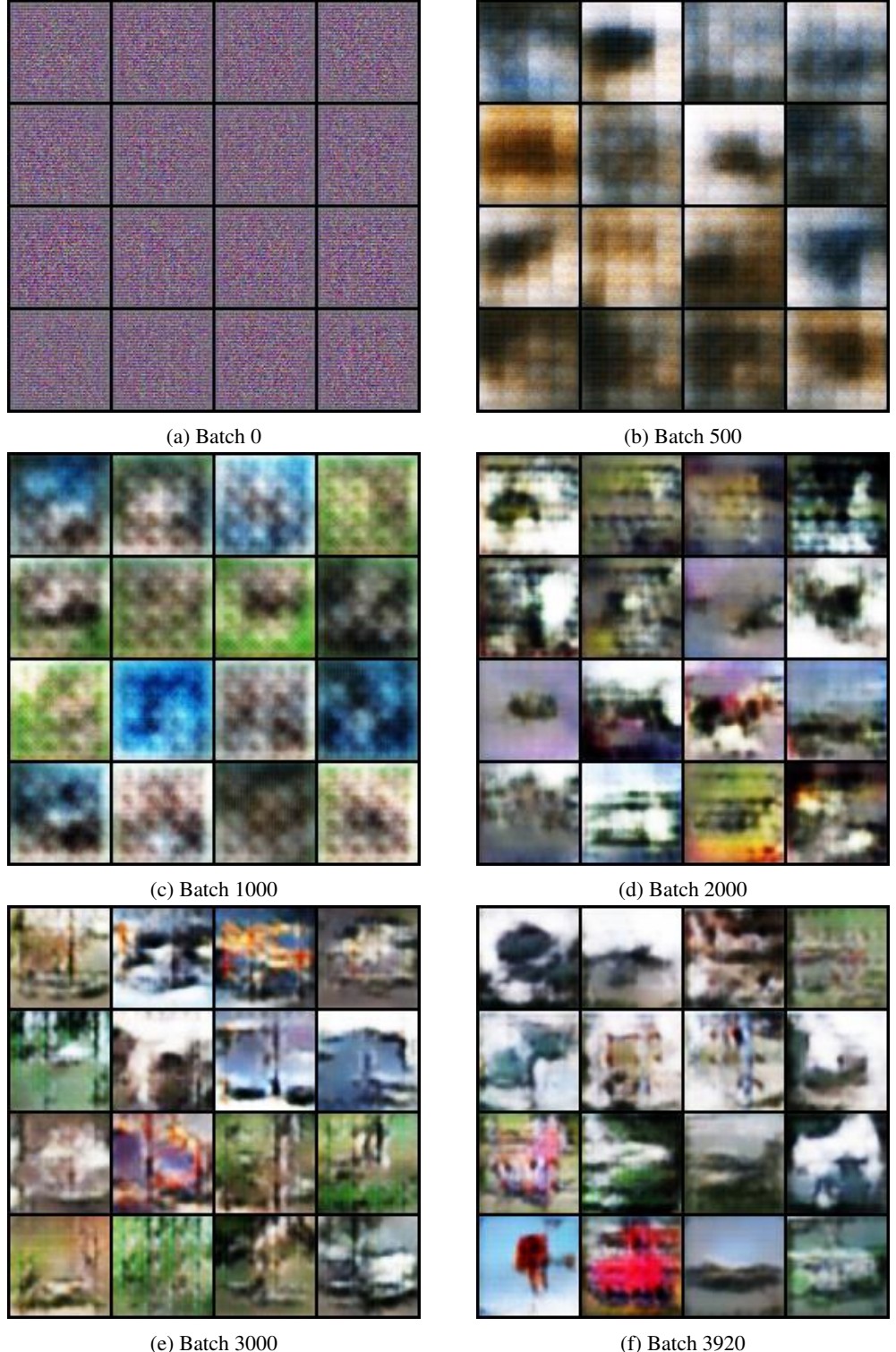

(a) Batch 0

(b) Batch 500

(c) Batch 1000

(d) Batch 2000

(e) Batch 3000

(f) Batch 3920

Figure 8: Samples from DCGAN trained on CIFAR-10 test set

## C    EFFECT OF BLURRING CIFAR-10

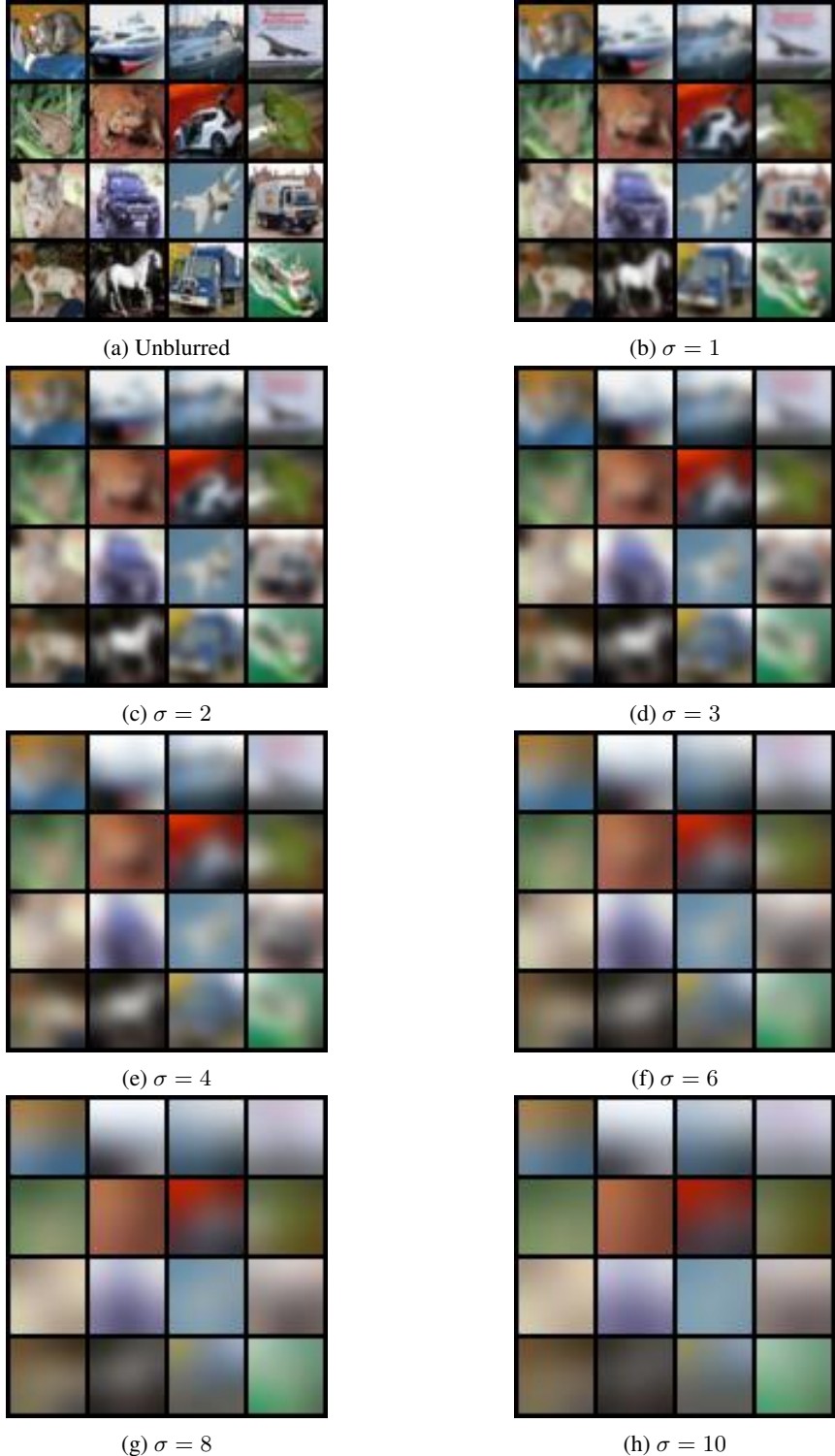

Figure 9: Effect of different $\sigma$ on $\beta_\sigma(X)$, for $X$ the CIFAR-10 training set

