# OpenReview forum: "Towards a Testable Notion of Generalization for Generative Adversarial Networks"
_ICLR.cc/2018/Conference — Reject_

### Official Review · AnonReviewer3 · 2017-11-27
**Interesting idea but realization could be improved**

**Rating:** 5
**Confidence:** 3

**Review:**

`The papers aims to provide a quality measure/test for GANs.   The objective is ambitious an deserve attention. As GANs are minimizing some f-divergence measure, the papers remarks that computing a  Wasserstein distance between two distributions made of a sum of Diracs is not a degenerate case and is tractable. So they propose evaluate the current approximation of a distribution learnt by a GAN by using this distance as a baseline performance (in terms of W distance and computed on a hold out dataset).

A first remark is that the papers does not clearly develop the interest of puting things a trying to reach a treshold of performance in W distance rather than just trying to minimize the desired f-divergence. More specifically as they assess the performance in terms of W distance I would would be tempted to just minimize the given criterion. This would be very interesting to have arguments on why being better than the "Dirac estimation" in terms of W2 distance would lead to better performance for others tasks (as other f-divergences or image generation).

According to the authors the core claims are:
"1/ We suggest a formalisation of the goal of GAN training (/generative modelling more broadly) in terms of divergence minimisation. This leads to a natural, testable notion of generalisation. "
Formalization in terms of divergence minimization is not new (see O. Bousquet & all https://arxiv.org/pdf/1701.02386.pdf ) and I do not feel like this paper actually performs any "test" (in a statistical sense). In my opinion the contribution is more about exhibiting a baseline which has to be defeated for any algorithm interesting is learning the distribution in terms of W2 distance.

"2/ We use this test to evaluate the success of GAN algorithms empirically, with the Wasserstein distance as our divergence."
Here the distance does not seems so good because the performance in generation does not seems to only be related to W2 distance. Nevertheless, there is interesting observations in the paper about the sensitivity of this metric to the bluring of pictures. I would enjoyed more digging in this direction. The authors proposes to solve this issue by relying to an embedded space where the L2 distance makes more sense for pictures (DenseNet). This is of course very reasonable but I would expect anyone working on distribution over picture to work with such embeddings. Here I'm not sure if this papers opens a new way to improve the embedding making use on non labelled data. One could think about allowing the weights of the embeddings to vary while f-divergence is minimized but this is not done in the submitted work.

 "3/ We find that whether our proposed test matches our intuitive sense of GAN quality depends heavily on the ground metric used for the Wasserstein distance."
This claim is highly biased by who is giving the "intuitive sense". It would be much better evaluated thought a mechanical turk test.

 "4/ We discuss how to use these insights to improve the design of WGANs more generally."
As our understanding of the GANs dynamics are very coarse, I feel this is not a good thing to claim that "doing xxx should improve things" without actually trying it.

---

> ### Author Response · Authors · 2018-01-05
> **Reply**
>
> Thanks a lot for your review.
>
> Our reason for not requiring D = D_\Gamma (and indeed for contradicting this in the case of the DCGAN, where D_\Gamma is the Jenson-Shannon divergence rather than the Wasserstein distance) is that we believe a GAN may still useful for minimising D even when D_\Gamma != D, due to the regularising effect that optimising over \mathcal{Q} entails. This is why we distinguish between the GAN objective (section 3), and our *overall* objective (section 2), which is what we ultimately care about. We definitely don't claim to be the first to talk about divergence minimisation in the context of GAN training, but we think that this distinction (between D and D_\Gamma) - as well as our direct treatment of the finiteness of our dataset, and our establishment of an intuitive performance baseline for generalisation - are useful contributions.
>
> We also believe that the uninituitive behaviour of W_L^2 (e.g. figure 2) is largely fixed by changing the ground metric as we describe. In this case, we obtain the much more plausible figure 5, where the value does appear to correspond to image quality. We agree that this is subjective, but think the result is still compelling. We are also not aware of any related Wasserstein GAN work in which the ground metric is defined in such a way, though welcome any such references.
>
> Finally, we certainly do not intend to claim that changing the ground metric will (or even should) improve GAN training. However, our results do suggest that this largely overlooked component of the WGAN is indeed significant, and our discussion simply aims to promote further consideration of this issue in a slightly more concrete way.
>
> Please also note that we have uploaded a revised copy of our paper which may clarify things further.

---

### Official Review · AnonReviewer2 · 2017-11-28
**Good paper, a bit wordy and lacking a few experiments**

**Rating:** 6
**Confidence:** 3

**Review:**

The quality of the paper is good, and clarity is mostly good. The proposed metric is interesting, but it is hard to judge the significance without more thorough experiments demonstrating that it works in practice.

Pros:
 - clear definitions of terms
 - overall outline of paper is good
 - novel metric

Cons
 - text is a bit over-wordy, and flow/meaning sometimes get lost. A strict editor would be helpful, because the underlying content is good
 - odd that your definition of generalization in GANs appears immediately preceding the section titled "Generalisation in GANs"
 - the paragraph at the end of the "Generalisation in GANs" section is confusing. I think this section and the previous ("The objective of unsupervised learning") could be combined, removing some repetition, adding some subtitles to improve clarity. This would cut down the text a bit to make space for more experiments.
 - why is your definition of generalization that the test set distance is strictly less than training set ? I would think this should be less-than-or-equal
 - there is a sentence that doesn't end at the top of p.3: "... the original GAN paper showed that [ends here]"
 - should state in the abstract what your "notion of generalization" for gans is, instead of being vague about it
 - more experiments showing a comparison of the proposed metric to others (e.g. inception score, Mturk assessments of sample quality, etc.) would be necessary to find the metric convincing
 - what is a "pushforward measure"? (p.2)
 - the related work section is well-written and interesting, but it's a bit odd to have it at the end. Earlier in the work (e.g. before experiments and discussion) would allow the comparison with MMD to inform the context of the introduction
 - there are some errors in figures that I think were all mentioned by previous commentators.

---

> ### Author Response · Authors · 2018-01-05
> **Reply**
>
> Thanks very much for your comments. We have uploaded a revised version of the paper that hopefully addresses many of the points that you making regarding the writing of the paper. Regarding your other points - we use a strict inequality in our condition because, if a GAN were merely *as good* as the training set, then it seems hard to justify all the effort in implementing it. (However, we would expect equality to hold with probability 0, so this is probably an edge case.) We also definitely agree that further experimental investigation is necessary, but we think that the implications of our findings about the Wasserstein GAN (namely, that we do not even get close to generalising - see Figure 5) and the significance of the ground metric (which has largely been overlooked) are still of interest to the community.

---

### Official Review · AnonReviewer1 · 2017-11-28

**Rating:** 4
**Confidence:** 4

**Review:**

This paper proposed a procedure for assessing the performance of GANs by re-considering the key of observation. And using the procedure to test and improve current version of GANs. It demonstrated some interesting stuff.

It is not easy to follow the main idea of the paper. The paper just told difference stories section by section. Based on my understanding, the claims are 1) the new formalization of the goal of GAN training and 2) using this test to evaluate the success of GAN algorithms empirically?  I suggested that the author should reform the structure, ignore some unrelated content and make the clear claims about the contributions on the introduction part.

Regarding the experimental part, it can not make strong support for all the claims. Figure 2 showed almost similar plots for all the varieties. Meanwhile, the results are performed on some specific model configurations (like ResNet) and settings. It is difficult to justify whether it can generalize to other cases. Some of the figures do not have the notations of curvey, making people hard to compare.

Therefore, I think the current version is not ready to be published. The author can make it stronger and consider next venue.

---

> ### Author Response · Authors · 2018-01-05
> **Reply**
>
> Thank you very much for your review. We have uploaded a revised version of the paper that significantly improves upon the issues of clarity you have mentioned. We hope this addresses some of the concerns you have raised. Regarding the experimental results: Figure 2 shows the results of applying our methodology to one specific case - the DCGAN trained on CIFAR-10 - so we would not expect different plots to vary significantly. We provided multiple runs to give an idea of how much variance there is in our method, but only one representative run is necessary to convey the result (and we have switched to the latter in the revised version). We agree that further experimental investigation is necessary, but we think that the implications of our findings about the Wasserstein GAN (namely, that we do not even get close to generalising - see Figure 5) and the significance of the ground metric (which has largely been overlooked) are still of interest to the community.

---

### Public Comment · (anonymous) · 2017-11-09
**about the proposed notion of generalization and other comments**

Hi, thanks for this interesting and easy to read paper. I have some questions and comments:

*Given your definition of generalization (eq. 1), one would infer that none of the GANs you tested generalize well, since in all figures the solid curves remain above the dashed line, is this correct? There is only one exception to this, that is Fig. 2 in which the solid curves do go below the dashed line, but then the quality of the samples is not good (Fig. 8). So it seems to me that eq. 1 is not a clean test for the performance of a GAN, and you need some extra, ad hoc procedure (as the one you propose with the trained neural network) to make sure it implies that the GAN is working. Would it be possible to have a general recipe that works for any type of data?

*I find very interesting the idea that focusing on eq.2, as Arora et al. 2017 proposed, might lead to degenerate choices of D. It is worth noting however that Arora and Zhang 2017 claim that "the Arora et al. scenario could involve the trained distribution having small support, and yet all its samples could be completely disjoint from the training samples". Thus it seems conceivable that a trained GAN could fulfill eq.1 but still being very far from the original distribution. Is this correct?

*Recently, Fedus et al. have also used the Wasserstein distance to assess the quality of GANs. They also state that only viewing GANs as minimizing a divergence is "overly restrictive", which might be worth mentioning in the context of your paper.

I have some other minor comments that I thought might help improving the paper:

* The sentence "For instance, the original GAN paper showed that" is not complete.

* Is the equation in page 5 correct? It does not seem to be consistent with its explanation: "blurring X by any amount brings...".

* Fig. 5 does not have a dashed line (although the line is mentioned in the caption). This dashed line is pretty relevant, since it should show that the proposed ground metric solves the issues seen in Fig. 2.

* Caption in Fig. 8 mentions I-WGAN when it corresponds to DC-GAN, right?

* Adding legends to figures could make them easier to understand.

* In some cases, it was difficult for me to follow the reasoning because there are variables defined but never used and some concepts referred with several letters.

* Typos (?):

  -We measure closeness measured in terms.
  - Note that on MNIST we modified first duplicated.
  - and alo for learning.
  - it metricizing weak.
  - sqaure-summable.


Hope all the above is clear and helps improving the paper. I think it is a well written paper and found very interesting some of the ideas discussed.

Thanks.

---

> ### Author Response · Authors · 2017-11-20
> **Response to comments**
>
> Hi, thanks a lot for your helpful suggestions and comments.
>
> Regarding your first few points:
>
> * We agree that Fig. 2 seems wrong. However, we think that the problem is not in (1) itself, but rather in (4), which is our empirical approximation of (1), and which Fig. 2 depicts. As the number of samples goes to infinity, (4) will converge to (1) almost surely (at least when D is a Wasserstein distance), but we think that, for Fig. 2, we simply didn't have enough samples for (4) to approximate (1) accurately yet. At present, we don't have results that indicate whether (4) will be an accurate approximation of (1) for a particular D given some budget of samples -- what we can do at present is run experiments and see if the results make sense, and then conjecture that this applies to other similar cases also.
>
> We also mention that we do not think changing the ground metric to the embedded L2 is particularly hacky, since the resulting D produced is still a valid Wasserstein distance (assuming the embedding is injective). In fact, we see the choice of L2 (which is almost always made implicitly) as somewhat arbitrary anyway -- especially for comparing images as in this context -- and believe that drawing attention to this issue is one of the main contributions of our work.
>
> * It is definitely possible that (1) might hold, and yet the generator might still be far from the true distribution (particularly when the training set is very small). However, we believe that (1) is still a useful condition to assess whether a GAN has at least done *something* -- namely, it shows whether or not we have gotten closer to the data distribution than we were a priori (when all we had was our training set). We still might have a long way to go to get to the data distribution, but in this case a good choice of D (such as a Wasserstein distance) would indicate this, since D(alpha(X), pi) would be large.
>
> * We completely agree that it is unclear whether GANs actually do minimise the divergences they claim to, especially given limited network capacity, and the method by which they are usually trained (alternating generator and discriminator steps) -- in fact, this work was largely motivated by a desire to test whether or not they do. To some extent, our model of a GAN in section 3 is unnecessary for our argument -- we could remain completely agnostic as to what exactly a GAN is doing, and simply consider it as a black box corresponding to one particular choice of alpha.
>
> However, we do see a divergence as necessary for formulating the overall problem that we are trying to solve with GANs (if not the mechanics of how a GAN will actually solve that problem). If our objective is to “learn the data distribution”, then we believe success or failure is naturally measured in terms of some divergence between the generator and the true distribution. We view the question of whether GANs strictly minimise this divergence at each training step as a separate (but related) question.
>
> For your remaining points:
>
> * Yes, the inequality on page 5 is the wrong way around.
>
> * The dashed line in Figure 5 was left out by mistake. We will fix this - in this case, the dashed line is a significant distance below the blue line (much like the graphs in Figure 4.)
>
> * Yes, Figure 8 corresponds to a DC-GAN. (A similar error occurred in Figures 2 and 5, which should refer to CIFAR-10 rather than MNIST.)
>
> Thank you very much also for those typos, and for your other suggestions.

---

### Public Comment · (anonymous) · 2017-11-10
**Related work**

I believe that the idea of evaluating the quality of a generator using a probability distribution divergence (on either pixels or pre-trained features) was firstly explored in [https://arxiv.org/abs/1705.05263] (neural network Wasserstein distance), [https://arxiv.org/abs/1610.06545] (neural network Jensen-Shannon), and [https://arxiv.org/abs/1708.04692] (both of them).

---

> ### Author Response · Authors · 2017-11-17
> **Differences from these works**
>
> Hi, thank you very much for your comment.
>
> It is certainly true that probability divergences have been previously suggested as a way to evaluate generator quality. However, as far as we are aware, this has not been used to formalise the notion of generalisation that we suggest, wherein the generator generalises if it moves closer to the data distribution than the empirical distribution of the training set.
>
> We also see several more specific differences between our work and the papers you mention.  To our understanding, Lopez-Paz & Oquab do not seek to minimise any divergence directly, but rather propose a two-sample test that aims to accept or reject the hypothesis that the generator and true distributions are identical. In particular, once a significance value has been chosen, the output of their test is simply a binary value indicating whether to accept or reject the hypothesis. Their proposed C2ST statistic does give a numerical value 0 <= t <= 1 that should give some information as to the "closeness" of the generator and true distributions, but this is not a statistical divergence as in our work.
>
> The other two papers you mention do evaluate generator quality using an approximate Wasserstein distance between empirical distributions. However, unlike us, they do so by training a GAN discriminator e.g. with the WGAN or WGAN-GP methods. As acknowledged in the original WGAN paper, the accuracy of this approach depends on significant assumptions about the class of functions attainable via a given discriminator architecture and training procedure. In contrast, we compute empirical Wasserstein distances exactly by solving a linear program, which requires no such assumptions.
>
> The use of a neural network embedding is also very different in these papers than in ours.  In particular, Danihelka et al. use an embedding only to speed up discriminator training, and not to evaluate generator quality as we do. On the other hand, Lopez-Paz & Oquab use an embedding to weaken the quality of real and generated samples, since otherwise these are easily distinguished with perfect accuracy by their binary classifier. In contrast, we use the embedding to change the ground metric for our Wasserstein distance, in the hope that speed up the convergence of the empirical Wasserstein distance to the true Wasserstein distance (and we suggest empirically that this does take place). In the process, we also reveal the significance of the ground metric, which we believe has largely been negelected in this field so far.

---

### Comment · AnonReviewer3 · 2017-11-22
**Some questions**


- I'd like the claims of the paper to be more clearly apparent. As an example it seems that the papers claims that using the (1) criterion is a good idea for testing the generalization quality of a generative model but fig 3 clearly shows that it is not always the case.

- In section (4) What is special about using a W distance in (1) to perform the GAN quality assertion?  Is it just because it can be estimated easily and is meaningful when the two distributions are a sum of Diracs?

-  There is a direct link between the cost optimized by a GAN and the minimization of f-divergences (see §2.1 of https://arxiv.org/pdf/1701.02386.pdf for details). Why would it not be enough to estimate this divergence on the test set ? In other word,  what is the interest to set a threshold on the divergence rather than just minimizing it?

- Is the idea proposed in §5 consist in solving the optimal transport problem in an embedding space (ResNet one as an example) rather than on the image space?

- About the form of the paper, I usually think that following the convention of ML papers and stating core claims as theorems whenever possible and highlighting proposed algorithms.

---

> ### Author Response · Authors · 2017-11-24
> **Response to questions**
>
> Hi, thank you for your 	questions and comments. We answer these in turn:
>
> * Our abstract and introduction lay out the main points of the paper. To give a quick summary here:
>   - We suggest a formalisation of the goal of GAN training (/generative modelling more broadly) in terms of divergence minimisation. This leads to a natural, testable notion of generalisation.
>   - We use this test to evaluate the success of GAN algorithms empirically, with the Wasserstein distance as our divergence.
>   - We find that whether our proposed test matches our intuitive sense of GAN quality depends heavily on the ground metric used for the Wasserstein distance.
>   - We discuss how to use these insights to improve the design of WGANs more generally.
>
> Regarding the figure (I assume you meant Fig. 2 rather than Fig. 3?): we think that the problem in Fig. 2 is not in (1) itself, but rather in (4), which is our empirical approximation of (1), and which Fig. 2 depicts. We believe that, if the terms in (1) were plotted instead, we would obtain a curve resembling Fig. 5. (Note that the dashed line on Fig. 5 was left out by mistake, but appears significantly below the line plotted, just like in Fig 4.)
>
> Now, as the number of samples goes to infinity, (4) will converge to (1) almost surely (at least when D is a Wasserstein distance), which we use to justify our empirical approximation. However, we think that in Fig. 2 we simply didn't have enough samples for (4) to approximate (1) accurately yet. We also believe that changing the ground metric to the embedded L2 improved the convergence rate, hence allowing us to achieve the more desirable Fig. 5 for the same experiment.
>
> We will clarify the distinction between (1) and (4) at greater length.
>
> * Our key reasons for choosing the Wasserstein distance are given in section 4. In particular, this choice makes D sensitive to the underlying topology of our data space (through the choice of ground metric). For example, the Wasserstein distance between two Dirac distributions varies continuously according to the distance between their masses, i.e.
>
> W_d(dirac_x, dirac_y) = d(x, y),
>
> as opposed to (say) KL(dirac_x, dirac_y), which is infinite if x != y and 0 otherwise. The Wasserstein distance also metricises weak convergence, allowing its approximation via the Wasserstein distance between empirical distributions (and it is also useful that this is tractable to compute, as you mention). This also means we make no density assumptions about the distributions involved -- all we need to compute this approximation is the ability to sample. (We will emphasise this last point more in the paper.)
>
> * We believe that the threshold (1) serves as a useful test of whether a GAN has generalised beyond its training data. As we say in section 2, if (1) holds, then “using alpha here actually achieved something: in a sense, it has injected additional information about pi into X_hat (perhaps through some sort of smoothing or regularisation), and brought us closer to pi than we already were a priori”.
>
> Otherwise put: we always have the option of choosing alpha(X) to be the empirical distribution of our dataset X_hat. If our aim is to choose a distribution alpha(X) that is as close (as measured by D) to the true distribution pi as possible, and if alpha(X) != X_hat, then (1) had better hold -- otherwise we could have done better by choosing alpha(X) = X_hat.
>
> * This question is incomplete - would you please clarify?
>
> * The proposal in section 5 essentially consists of training a WGAN in the usual way, but with the discriminator given by h(eta(x)), where h is learned but eta is a fixed embedding. We justify in that section why this corresponds to changing the ground metric from the standard L2 to the eta-embedded L2. We will clarify this further.
>
> * Yes, we agree this would be useful (particularly highlighting the proposed algorithm) and will modify the paper paper accordingly.
>
> Thanks again for your input.

---

### Decision · Program_Chairs · 2018-01-29
**ICLR 2018 Conference Acceptance Decision**

**Decision:**

Reject

**Comment:**

This paper proposes a method for quantitatively evaluating GANs. Better quantitative metrics for GANs are badly needed, as the field is being held back by excessive focus on generated samples. This paper proposes to estimate the Wasserstein distance to the data distribution. A paper which does this well would be a significant contribution, but unfortunately (as the reviewers point out) the experimental validation in this paper seems insufficient.

To be convincing, a paper would first need to demonstrate the ability to accurately estimate Wasserstein distance -- not an easy task, but one which receives little mention in this paper. Then it would need to validate that the method can either quantitatively confirm known results about GANs or uncover previously unknown phenomena. As it stands, I don't think this submission is ready for publication in ICLR, but I'd encourage resubmission after more careful experimental validation along the lines suggested by the reviewers.